# Effects of Low-Level Laser Therapy and Purified Natural Latex (*Hevea brasiliensis*) Protein on Injured Sciatic Nerve in Rodents: Morpho-Functional Analysis

**DOI:** 10.3390/ijms241814031

**Published:** 2023-09-13

**Authors:** Fernando José Dias, Diego Pulzatto Cury, Paula Elisa Dias, Eduardo Borie, Josefa Alarcón-Apablaza, María Florencia Lezcano, Paulina Martínez-Rodríguez, Daniel Vargas, Brandon Gutiérrez, Valéria Paula Sassoli Fazan

**Affiliations:** 1Oral Biology Research Centre (CIBO-UFRO), Dental School—Facultad de Odontología, Universidad de La Frontera, Temuco 4780000, Chile; paulinaconstanza.martinez@ufrontera.cl; 2Department of Integral Adults Dentistry, Dental School—Facultad de Odontología, Universidad de La Frontera, Temuco 4780000, Chile; eduardo.borie@ufrontera.cl; 3Department of Anatomy and Department of Cellular Biology and the Development—Institute of Biomedical Sciences, Universidade de São Paulo (ICB-USP), São Paulo 05508-000, Brazil; diegocury@usp.br; 4School of Pharmaceutical Sciences of Ribeirao Preto, Universidade de São Paulo (FCFRP-USP), Ribeirão Preto 14040-903, Brazil; paulaelisadias@gmail.com; 5Research Centre in Dental Sciences (CICO-UFRO), Dental School—Facultad de Odontología, Universidad de La Frontera, Temuco 4780000, Chile; josefa.alarcon@ufrontera.cl; 6Doctoral Program in Morphological Sciences, Medical School, Universidad de La Frontera, Temuco 4780000, Chile; daniel.vargas@umayor.cl; 7Departamento de Bioingeniería, Facultad de Ingeniería, Universidad Nacional de Entre Ríos, Oro Verde 3100, Argentina; lezcano.f@gmail.com; 8Master Program in Dental Sciences, Dental School, Universidad de La Frontera, Temuco 4780000, Chile; brandon.gutierrez@ufrontera.cl; 9Department of Surgery and Anatomy, School of Medicine of Ribeirão Preto, Universidade de São Paulo (FMRP-USP), 14049-900 Ribeirão Preto, Brazil

**Keywords:** nerve regeneration, LLLT, F1 protein, functional analysis, morphometry, ultrastructure

## Abstract

The present study analyzed the effects of low-level laser therapy (LLLT) and the purified natural latex protein (*Hevea brasiliensis*, F1 protein) on the morpho-function of sciatic nerve crush injuries in rats. One-hundred and eight male Wistar rats were randomly allocated to six groups (*n* = 18): 1. Control; 2. Exposed (nerve exposed); 3. Injury (injured nerve without treatment); 4. LLLT (injured nerve irradiated with LLLT (15 J/cm^2^, 780 nm)); 5. F1 (injured nerve treated with F1 protein (0.1%)); and 6. LLLT + F1 (injured nerve treated with LLLT and F1). On the 1st, 7th, 14th, and 56th days after injury, a functional sensory analysis of mechanical allodynia and mechanical hyperalgesia and a motor analysis of grip strength and gait were performed. After 3, 15, and 57 days, the animals were euthanized for morphometric/ultrastructural analyses. The treatments applied revealed improvements in morphometric/ultrastructural parameters compared to the injured group. Sensory analyses suggested that the improvements observed were associated with time progression and not influenced by the treatments. Motor analyses revealed significant improvements in grip strength from the 7th day in the LLLT group and in gait from the 56th day in all treated groups. We concluded that even though the morphological analyses showed improvements with the treatments, they did not influence sensory recovery, and LLLT improved motor recovery.

## 1. Introduction

The high incidence of peripheral nerve lesions has aroused the interest of researchers, which has instigated pre-clinical and clinical research on nerve regeneration [1]. Although there is some degree of self-recovery in nerve injuries, the process is slow, and often incomplete, resulting in the loss or diminution of motor, sensory, and autonomic functions [2].

Crush injury is an experimental induced nerve injury that results in axonal disruption, with the preservation of connective sheaths, resulting in an axonotmesis-type injury [3,4]. The difficulty, in this case, lies in the variety of forms of nerve crushing, which can be reduced by standardizing the injury [5].

The use of low-level laser therapy (LLLT) is an interdisciplinary approach to nerve regeneration [6,7]. In recent years, it has become an increasingly important modality and includes peripheral nerve regeneration, whose exact mechanisms of action of the therapeutic laser are not yet fully understood [8]. However, studies reveal that it can induce trophic conditions and inhibition of the inflammatory process necessary for nerve regeneration [9]. In addition, it maintains the functional activity of the injured nerve for a prolonged period, decreases scar tissue formation at the injury site, and significantly increases axonal growth and myelination [10].

In the 1990s, at the Neurochemistry Laboratory (Ribeirão Preto Medical School, University of São Paulo), a new biocompatible material extracted from *Hevea brasiliensis* (rubber tree) was developed and called the F1 protein [11]. Initially, this biomaterial was used as a natural latex biomembrane, achieving promising regenerative results [12,13,14,15,16]. However, the use of this biomembrane for the regeneration of peripheral nerves has the drawback of not being biodegradable [17]. Thus, the F1 protein in absorbable form has shown advantages in some cases [18,19,20,21,22]. Natural latex serum showed an angiogenic effect and was effective in increasing vascular permeability [11,14,15,16]. Although it showed promising results, the absorbable form of the F1 protein has not yet been extensively investigated in nerve regeneration.

Several studies have shown positive results both in the use of low-level laser therapy and natural latex (*Hevea brasiliensis*) in the recovery of various types of tissues, including peripheral nerves; however, the literature is weak on this subject, presenting contradictory results and reports on the possible effects of combinations of techniques for the treatment of peripheral nerve injuries [23].

Gathering the presented information, the present report aims to analyze the effect of LLLT associated with the purified natural latex protein (F1 protein) on controlled compression injury (axonotmesis) in the sciatic nerve of Wistar rats through motor and sensory function tests, associated with the morphological, morphometric, and ultrastructural analysis of the nerves.

## 2. Results

### 2.1. Morphological Analysis

Our morphological analysis revealed a clear reduction in the density and diameter of the nerve fibers in the injured groups (Injury, LLLT, F1, and LLLT + F1) 3 days after the injury compared to the Control and Exposed groups. On days 15 and 57 post-injury, the injured groups showed morphological improvements, but still had inferior characteristics to those observed in the non-injured groups. The Injury group had a large area without the presence of nerve fibers at 15 days; however, at 57 days, this group revealed improvement and similar characteristics to the LLLT group. The F1 and LLLT + F1 groups were more similar to the uninjured groups, but still showed a decrease in diameter at 57 days (Figure 1).

### 2.2. Quantitative and Morphometric Analysis

#### 2.2.1. Nerve Fiber Density

Three days after nerve injury, the Control and Exposed groups revealed the highest values of this parameter. Among the treated groups, it was observed that the LLLT, F1, and LLLT + F1 groups were equal to each other and had higher values than the Injury group.

After 15 days of injury, the Control and Exposed groups had the highest nerve fiber densities. The treated groups were similar to each other, revealing an increase in nerve fiber density compared to the previous period. The LLLT and LLLT + F1 groups showed similar densities to the control group, and the density of the F1 group was similar to that of the Exposed group. The Injury group continued to show significantly lower values of nerve fiber density.

After 57 days of nerve injury, no differences were observed between the study groups concerning nerve fiber density (Figure 2).

The intragroup analysis in the different periods revealed a significant increase in nerve fiber density, mainly in the groups submitted to nerve injury (Injury, LLLT, F1, and LLLT + F1) between 3 and 15 days. The injury group also showed an increase in nerve fiber density between the periods of 15 and 57 days.

#### 2.2.2. Minimum Diameter of Nerve Fibers

The analysis of the minimum diameter of the nerve fibers after 3 days of injury revealed that the Control, Exposed, and F1 groups were similar and had the highest values of this parameter. The LLLT + F1 group was similar to the Control group. The lowest values of minimum diameter in this period were observed in the LLLT and Injury groups, respectively.

At 15 days after the lesion, the highest diameter values were observed in the Control and F1 groups, followed by the Exposed group. The Injury, LLLT, and LLLT + F1 groups were similar, presenting the lowest values in this period.

Finally, 57 days after the lesion, the Control and Exposed groups presented the largest diameters. This was followed by the treated groups F1, LLLT + F1, and LLLT, and the Injury group, which revealed the smallest diameters of nerve fibers (Figure 3).

Upon comparing the same groups in the different periods of analysis, even though the groups revealed significant differences, a general pattern of modification in the diameters of nerve fibers was not noted. It was only observed that the diameter of nerve fibers increased in the Control and Exposed groups in the period of 57 days.

#### 2.2.3. Blood Capillary Density

Regarding the density of blood vessels at 3 days, the Control and Exposed groups were similar, presenting the highest values of capillary density. All lesioned groups were similar, with lower blood vessel density.

The density of the blood vessels at 15 and 57 days of lesions did not reveal significant differences between the study groups, and an increasing trend was noted between the lesioned groups with or without treatment (Figure 4).

The analysis of the same groups in different periods revealed that the density of blood capillaries was reduced in the control group after 57 days and in the injured and treated groups (LLLT, F1, and LLLT + F1) compared to the 3-day period.

### 2.3. Ultrastructural Analysis—Transmission Electron Microscopy

In the ultrastructural analysis, it was possible to observe in the Control and Exposed groups after 3, 15, and 57 days, after carrying out the experimental protocol, that the myelinated and unmyelinated nerve fibers presented a normal appearance, were well distributed throughout the cross-sectional area of the sciatic nerve, in amidst blood capillaries and Schwann cells with an evident nucleus. No abnormal structures or signs of nerve degeneration were noted in these groups (Figure 5A,B).

The groups submitted to nerve injury (Injury, LLLT, F1, and LLLT + F1) revealed, at 3 days after nerve injury, extensive areas without nerve fibers and areas of Wallerian degeneration with axons and Schwann cells in the process of degeneration. In the F1 and LLLT + F1 groups, cleaner areas with few axons and more intact Schwann cells were noted, especially in comparison with the injury group (Figure 5C–F).

After 15 days of nerve injury, a level of recovery was observed in all injured groups, with a greater number of Schwann cells, better distributed nerve fibers with a smaller diameter, and a reduced thickness of the myelin sheath compared to the 3-day period. The presence of clusters of a few nerve fibers due to Schwann cells was noted (Figure 5G–J).

On the 57th day, the samples from the injured groups again showed improvements compared to the previous periods. A greater number of nerve fibers with thicker myelin sheaths were well distributed throughout the cross-sectional area of the nerve. In addition to many Schwann cells, the presence of blood capillaries was noted. However, smaller diameters of the axons were noted and, although they were less evident, groupings of nerve fibers by Schwann cells were still observed (Figure 5K–N).

### 2.4. Sensory Function Evaluation

#### 2.4.1. Mechanical Allodynia

The mechanical allodynia test after one of the nerve injuries revealed significantly higher values in the injured groups (Injury; LLLT; F1 and LLLT + F1) that were similar to each other than the uninjured groups (Control and Exposed). In the other analyzed periods, two values of applied load remained in all the groups subjected to nerve injury. After 7, 14, and 56 days of injury, no significant differences were observed between the groups. However, a tendency toward reduction in two values was observed after 7 days of nerve lesion in the F1 group, approaching the values of the Control and Exposed groups. The Injury group showed higher values after 14 and 56 days of nerve injury (Figure 6).

The intragroup analysis in the different periods showed that the non-injured animals (Control and Exposed) did not reveal differences in mechanical allodynia among the analyzed periods. The injured groups (Injury, LLLT, F1, LLLT + F1) showed a significant reduction in values after the first day, which remained similar after 7, 14, and 56 days of nerve injury.

#### 2.4.2. Mechanical Hyperalgesia

The analysis of hyperalgesia showed significantly higher charges for a positive response in the injured groups (Injury, LLLT, F1, and LLLT + F1) on the first day after injury compared to the Control and Exposed groups. The injured groups showed a gradual reduction in load starting from the 7-day period and the uninjured groups revealed an increase in nociceptive threshold for the response. From the 7th day after nerve injury, no significant differences were observed between the study groups. Therefore, attention is drawn to the sudden reduction at 7 days in the LLLT group and at 14 days after injury in the LLLT + F1 group (Figure 7).

A comparison of the same groups in different periods also showed that non-injured animals (Control and Exposed) did not reveal differences in mechanical hyperalgesia among the analyzed periods. Again, the injured groups (Injury, LLLT, F1, LLLT + F1) showed a significant reduction in values after the first day, which remained similar after 7, 14, and 56 days of nerve injury.

### 2.5. Motor Function Evaluation

#### 2.5.1. Grip Strength Test

It was possible to notice an increase from less than 50 g to almost 300 g in average grip strength in all groups during the 56 days evaluated.

After 1 day of the nerve injury, the grip strength of two injured groups was significantly lower than that of the control group. After 7 days, the LLLT and LLLT + F1 groups revealed similar values to the uninjured groups. On the 14th day after the injury, all the injured groups presented similar values to the exposed group. And finally, after 56 days, no significant differences were observed between all the study groups (Figure 8).

The analysis of grip strength of the same groups in different periods generally revealed a gradual and significant increase in values in all treatment protocols from day 1 to day 56.

#### 2.5.2. Gait Assessment (Sciatic Nerve Functional Index)

The analysis of the gait of two animals carried out via SFI showed that in the periods of 1, 7, and 14 days after injury, the results of the two Control and Exposed groups were similar and significantly better than those of the injured groups. As time progressed, the injured groups were similar from the 1st to the 14th day of analysis.

After 56 days of nerve injury, improvement was observed in all the injured groups. The LLLT, F1, and LLLT + F1 groups presented similar values to the Control and Exposed groups, and the Injury group still showed significantly lower values (Figure 9).

The intragroup analysis of gait in the different periods, measured using the sciatic functional index, revealed that in the animals not submitted to nerve injury (Control and Exposed), there were no significant differences among the evaluated periods. The injured groups (Injury, LLLT, F1, and LLLT + F1) showed similarities in their SFI values after 1, 7, and 14 days of nerve injury, with these parameters improving only on day 56.

## 3. Discussion

The present report evaluated the morpho-functional characteristics of recovery from sciatic nerve crush injury after the application of treatment protocols with LLLT and purified natural latex protein (F1), complementing the analyses of previous studies by our group of researchers [18,19,20,21,22]. In general, improvements were observed in the morphological aspects and parameters of the crushed sciatic nerve with the different treatments proposed in the analyzed periods, and it was not possible to recognize a better treatment protocol. The sensory function results did not reflect the improvement observed in the morphological analyses; only the gait analysis was congruent with the morphological results.

The nerve lesion inflicted in this study did not spontaneously recover after 56 days. A load of 15 kgf (150 N) on the sciatic nerve was chosen due to its capacity for axonal destruction [24] and demyelination of the fibers [25]; this allowed us to evaluate recovery through the use of LLLT and the F1 Protein.

After 56 days, the morphological parameters of the Injury group were not similar to those of the non-injured groups, confirming the report by Muratori et al. [26]. Despite the injury inflicted, the treatment protocols showed a certain level of improvement in nerve fiber density with similar characteristics to the non-injured groups, but with smaller diameter fibers.

Our morphological/morphometric data showed an increase in the nerve fiber density, fiber diameter, and vascularization of the sciatic nerve in the injured and treated groups over time. The application of LLLT and the latex F1 protein showed morphological improvement in the first two weeks. The F1 group showed results closer to those of the non-injured groups 15 days after the injury. Finally, the impacts on the density of blood capillaries of the application of the proposed treatment protocols were not statistically confirmed. The morphometric parameters observed in different periods reinforced that the spontaneous recovery of nerve injury was time-dependent.

Our ultrastructural analysis corroborates our morphometric observations, confirming the reduction in the amount and size of nerve fibers after injury and the improvement in recovery characteristics with advancing time. Regarding the analysis of unmyelinated nerve fibers, there was an increased number of Schwann cells and grouping of nerve fibers, which is an expected characteristic in the recovery of crush injury [18,21,22,27,28].

An important advancement in our experimental protocol compared to previous studies [18,19,20,21,22] was to include sensory and motor function analysis to broaden the understanding of the action of LLLT and the F1 protein in nerve injury. This allowed us to analyze function recovery, which is a major objective of therapies for peripheral nerve injuries [29,30].

Some studies have reported an improvement in allodynia and hyperalgesia with the use of LLLT. These studies have already demonstrated an improvement in the peripheral nociception of the inferior alveolar nerve (904 nm, 9500 Hz—pulsed, 6 J/cm^2^) [31]; in transcranial application (810 nm, 7.2 J/cm^2^) [32]; in oxaliplatin-induced acute peripheral neuropathy (7.5 J/cm^2^, GaAlAs, 780 nm) [33]; in peripheral nerve damage in streptozotocin-induced type 1 diabetes (GaAr, 9500 Hz-pulsed, 6.23 J/cm^2^) [34]; and in the case of hyperesthesia induced through the injection of 5% formalin (830 nm, 8 J/cm^2^) [23]. No studies have evaluated mechanical allodynia and hyperalgesia after the application of the F1 protein to nerve injuries.

Our results of mechanical allodynia and hyperalgesia showed worse values on the first post-injury day in animals subjected to nerve injury. From the 7th to the 56th day, the values of all groups, injured or not, were similar, even with high variability. The passing of time made these data closer among the groups. Therefore, it is suggested that the treatments applied did not reflect on the improvement in sensory recovery, in which case, time and spontaneous recovery were the factors that influenced the recovery of nerve injury.

Regarding the motor assessment, the grip strength test most clearly reveals the effects of loss of axonal function [35]. Our results revealed that from day 1 to 56, there was an increase in force values, regardless of the protocol applied, suggesting that this increase may have occurred due to the increase in strength related to the growth of animals. In addition, there was a significant improvement in the LLLT and LLLT + F1 groups on the 7th day after nerve injury, with values similar to those of the control group. This improvement was not observed in the F1 group during this period, suggesting that the preponderant factor for this improvement was the application of LLLT. However, this improvement was maintained in the LLLT and LLLT + F1 groups after 15 days.

Our results corroborated other findings where manual grip strength improved significantly with the application of LLLT (3 J/cm^2^) [36], and five weekly applications of LLLT (0.8 J/cm^2^) in patients with ulnar nerve compression at the elbow [37].

These studies show that the performance of the LLLT can improve grip strength in patients, as observed in a present study with rats. However, there are studies that do not corroborate this improvement obtained using LLLT. Paolillo et al. [38] demonstrated that the weekly application of LLLT (18 J, 10 mW) in patients with osteoarthritis did not improve grip strength; however, in this case, the application had a low frequency compared to that of Ozkan et al. [37], which irradiated a large amount of energy of 18 J. In the present study, the energy deposited per point was 0.6 J, totaling 1.8 J in the three-points-per-application session.

Once more, no studies were found using the F1 protein that analyze its effect on grip strength. Our results did not reveal improvements in this motor function in any period analyzed in the comparison of the F1 group to the injury group.

Finally, our gait analysis suggested that all injured groups, treated or not, revealed a significant impairment of gait after the injury that persisted until the 56th day. On the 56th day, all the groups treated with LLLT and/or F1 presented significant improvements in gait compared to the injured group, being similar to the non-injured groups.

Our results corroborate many studies [39,40,41,42,43,44,45,46] that report an improvement in gait after the application of LLLT in crush injury of the sciatic nerve. However, different parameters were used concerning the wavelength (660, 685, 808, 830, and 1064 nm) and mainly the energy density (3, 4, 10, 35, 40, 50, 70, 80, 133, 140, and 280 J/cm^2^). This reveals a lack of uniformity in LLLT irradiation protocols. Thus, we are not able to point out which would be the most suitable parameters for this type of nerve injury.

The only study that previously analyzed the effect of the F1 protein on gait after nerve injury used a non-resorbable tubular scaffold containing the F1 protein and also reported gait improvements [47].

Thus, our results suggest that both separate and combined treatments, using LLLT and the F1 protein, resulted in similar improvements in gait in the studied animals, without evidencing possible potentiation of the results due to the combination of treatments.

Among the limitations of the study, we can mention the lack of morphological and ultrastructural analysis after 7 days of injury, as was with the case for the functional analyses, in addition to the lack of analysis more than 56 days after the nerve injury; the choice not to carry out these analyses in the periods mentioned was due to the reduction in the number of animals. Furthermore, this study used morphological/morphometric, ultrastructural, and functional methods without performing other types of biological analyses, for example, the analysis of inflammatory factors, which could influenced the results obtained in the present study.

## 4. Materials and Methods

This study used a methodology similar to those used in other previous studies by our research group [18,19,20,21,22], with a focus on adding the analysis of sensory and motor function analysis methods to the morphological methods of peripheral nerve regeneration of low-level laser treatment (LLLT) and the natural latex protein F1, with the intention of improving the understanding of the association of these two treatment modalities.

### 4.1. Animals

One-hundred and eight male Wistar rats (200–250 g) of two months of age were used, and were randomly distributed into 6 experimental groups (*n* = 18). Four animals were housed in polypropylene boxes with food and water provided *ad libitum*, in a temperature-controlled environment (22–24 °C), with 12 h cycles of daily lighting and air changes. This study was approved by the Local Scientific Ethics Committee of the of the Universidad de La Frontera (approval number 101_19), in accordance with the ethical principles of international animal experimentation.

### 4.2. Surgical Procedure and Nerve Injury

The animals were anesthetized with ketamine (75 mg/kg) and xylazine (10 mg/kg) and subsequently subjected to asepsis and trichotomy of the left hindlimb. The sciatic nerve was exposed, isolated, and crushed in a controlled manner with a load of 150 N (15 kgf) for a period of 10 min via deadlift [24] using equipment developed at the University of Sao Paulo (Ribeirao Preto, Brazil). Subsequently, the nerve was repositioned, and the skin was sutured with a 4–0 nylon thread, followed by the administration of anti-inflammatory 0.1 mL/100 g (Flunixin meglumine—Banamine, MSD animal health—São Paulo, Brazil) and a broad-spectrum antibiotic (Pentabiotic, Fort Dodge—Campinas, Brazil).

### 4.3. Low-Level Laser Therapy (LLLT)

For the LLLT irradiation, a “Twin Laser” device (MMOptics, São Carlos, Brazil) was used. Six irradiations were performed on alternate days in the proximal, middle, and distal regions of the injured sciatic nerve. The irradiation parameters follow the data in Table 1.

### 4.4. Application of the Purified Protein of Natural Latex (Hevea brasiliensis)

The 80% purified protein from natural latex at a concentration of 0.1% (*Hevea brasiliensis*) was mixed with hyaluronic acid at a concentration of 1% isolated from Gram-negative bacteria [48]. Both were mixed and filtered into sterile culture flasks (0.22 µm Millipore filters). The mixture was deposited in the nerve lesion prior to the suturing procedure.

### 4.5. Experimental Groups

Descriptions of the experimental protocols applied are presented in Table 2.

All animals were anesthetized and submitted to trichotomy in the left hind paw (pelvic limb). All groups underwent sensory and motor evaluation on days 1, 7, 14, and 56 after nerve injury. Each of these 6 groups was subdivided into 3 sub-groups of 6 animals, euthanized on days 3, 15, and 57 for histological analysis.

### 4.6. Sample Processing—Histological and Ultrastructural Analysis

After 3, 15, and 57 days of nerve injury, the animals were anesthetized (ketamine 75 mg/kg and xylazine 10 mg/kg) and euthanized via intracardiac perfusion (buffered saline solution (PBS 0.05 M, pH 7.4) followed by fixative solution (2.5% glutaraldehyde in 0.1 M sodium cacodylate buffer, pH 7.2). The sciatic nerves were then removed and kept submerged for 12 h at 4 °C in the same fixative solution as that used in the perfusion. The nerves were numbered, and washed in 0.1 M sodium cacodylate buffer, pH 7.2, for 1 h; then, they were post-fixed in 4% aqueous osmium tetroxide (OsO_4_), 0.2 M sodium cacodylate buffer, pH 7.2 and 6% aqueous potassium ferrocyanide (K_4_Fe(CN)_6_·3H_2_O) in a 1:2:1 ratio, for 2 h at room temperature, and then, 3 more times for 20 min (0.1 M cacodylate buffer), and were dehydrated in increasing concentrations of ethanol (25, 35, 50, 70, 75, 80, 95%). Then, they were infiltrated once in resin and ethanol in a 1:2 ratio for 2 h, and a second time in the inverse proportion. Finally, the fragments were infiltrated in complete resin with subsequent polymerization.

Cross sections 0.2 to 0.3 µm thick were obtained using an MT 6000-XL ultramicrotome (RMC Boeckeler, Tucson, AZ, USA). They were stained with 1% toluidine blue in saturated boric acid. These sections were used for light microscopy studies [49,50,51,52,53,54].

Ultrathin cross sections (50 to 60 nm) were obtained using diamond knives (RMC, Diamond Knives, Nidau, Switzerland), mounted on copper grids (slot, single hole) coated with 5% formvar, and contrasted with lead citrate and 6% uranyl acetate in an alcohol solution [55,56,57,58]. These sections were used for transmission electron microscopy analysis using a JEOL 1010 (Akishima, Japan) from the Institute of Biomedical Sciences of the University of São Paulo.

### 4.7. Morphometric and Quantitative Histological Analysis

After staining with toluidine blue, the histological sections were photographed using an M2 microscope equipped with an AxioCam MRc camera (Zeiss, Oberkochen, Germany). The densities of nerve fibers (nerve fibers/1000 μm^2^), blood capillaries (capillaries/mm^2^), and the minimum diameter of nerve fiber were calculated using ImageJ 1.53e software (National Institutes of Health, Stapleton, NY, USA). These data were compared between the different study groups, in the different periods analyzed.

### 4.8. Sensory Function Assessment

Mechanical allodynia: This was analyzed by applying von Frey filaments [59,60,61], with values of 0.05 g, 0.2 g, 2 g, 4 g, 10 g, and 300 g, in increasing order of filaments in the plantar regions of both hind paws, for 8 s. The filament that elicited two consecutive withdrawal responses was considered the stiffness necessary to induce the response (Figure 10A).

Mechanical hyperalgesia: This was assessed using a paw pressure test, on the two hind paws separately [63], using the Randall–Selitto device (Insight, Ribeirão Preto, Brazil). During the test, the examiner gradually increased the compression force until a withdrawal reflex occurred. The value recorded on the device was considered for data analysis [64] (Figure 10B).

### 4.9. Motor Function Assessment

Grip strength test: This was assessed using the “Grip Strength Meter” (GSM) device (Insight, Ribeirao Preto, Brazil) on both hind legs at the same time. The animals were encouraged to grasp the bar of the device with their hind legs. The examiner then exerted a small force, pushing the animal away from the device [65]. The tests were performed at all experimental times, three times, with intervals of 2 s between them. The grip strength was determined as an average of the forces recorded on the GSM in the 3 consecutive tests (Figure 10C).

Gait Assessment (Sciatic Nerve Functional Index): The “Walk-test” (Insight, Ribeirao Preto, Brazil) device was used to record the animals and obtain measurements of their gait. The parameters measured were “print length” (PL); “full extension” (TS); and “intermediate fingers” (IT) on the uninjured side (normal, right—NPL, NTS, and NIT) and on the injured side (experimental, left—EPL, ETS, and EIT). To calculate the sciatic functional index (SFI), the following calculation must be performed [66] (Figure 10D,E).
$$SFI=73\times \left[\left(\;\frac{NPL-EPL}{EPL}\;\right)+\left(\;\frac{ETS-NTS}{NTS}\;\right)+\left(\;\frac{EIT-NIT}{NIT}\;\right)\;\right]$$

### 4.10. Statistical Analysis

Statistical analysis of the data was carried out using SigmaPlot 12.0 software (Systat Software Inc. San Jose, CA, USA). The two-way ANOVA statistical test (analysis of variance, *p* = 0.05) was adopted with the Holm–Sidak post-test for multiple comparisons, and the data were presented as mean ± standard deviation.

## 5. Conclusions

In conclusion, our results revealed an improvement in the morphological/morphometric parameters of the treated groups in comparison to the injured group, however, without reaching similar values to the non-injured groups. The sensory function analyses revealed worse results after one day of nerve injury and similar values in all of the other analyzed periods, suggesting a lack of effectiveness of the treatments applied for recovery. The motor function analyses suggested an improvement in grip strength in the LLLT groups from the 7th day after injury. Gait showed significant improvement in all groups treated after 56 days of injury, with no differences among the treatments.

Gathering all the results, the morphological/morphometric observations showed some correlation with the functional data. However, the proposed treatment protocols were not effective for sensory recovery, and LLLT was efficient in motor recovery. Further studies considering different biological methods and LLLT and F1 protein treatment protocols will be necessary to confirm the results of the present report.

## Figures and Tables

**Figure 1 ijms-24-14031-f001:**
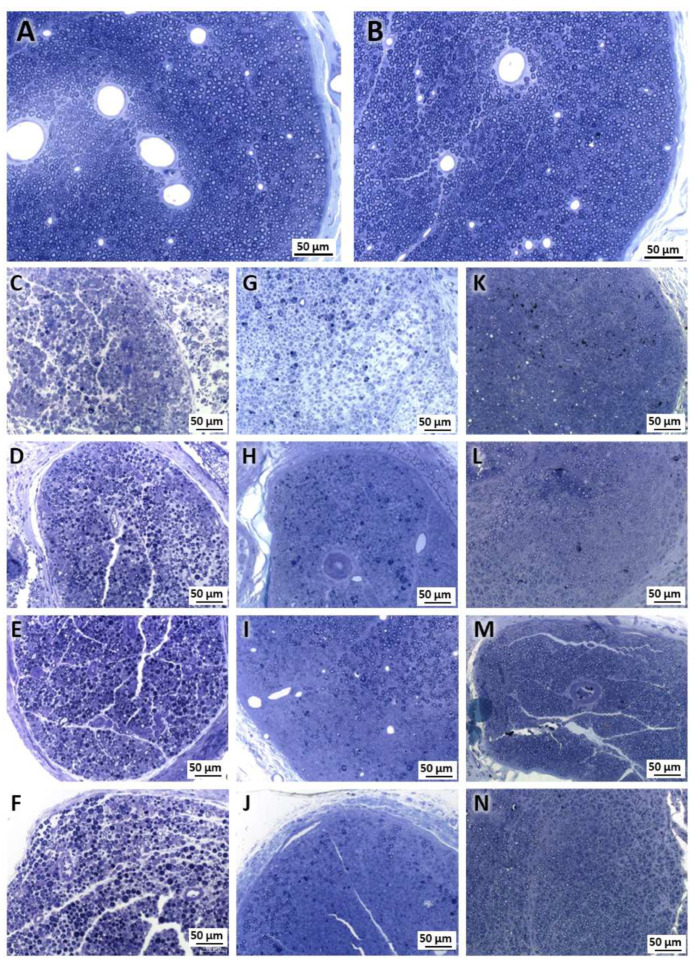
Morphological characteristics. (**A**) Control group (15 days) and (**B**) Exposed group (15 days), revealing normal nerve tissue 3 days after nerve injury. (**C**) Injury group. (**D**) LLLT group. (**E**) F1 group and (**F**) LLLT + F1 group, revealing reduced nerve fibers and degeneration regions 15 days after nerve injury. (**G**) Injury group. (**H**) LLLT group. (**I**) F1 group and (**J**) LLLT + F1 group, revealing some level of recovery compared to previous period 57 days after nerve injury. (**K**) Injury group. (**L**) LLLT group. (**M**) F1 group and (**N**) LLLT + F1 group, showing a higher number of nerve fibers.

**Figure 2 ijms-24-14031-f002:**
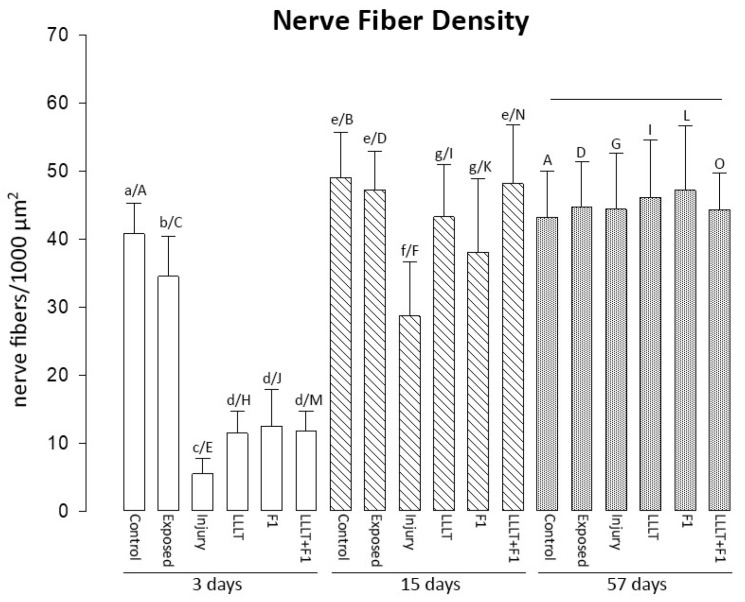
Nerve fiber density after 3, 15, and 57 days of nerve injury. The different lower-case letters (3 days: a, b, c, and d; 15 days: e, f, and g) represent significant differences (*p* < 0.05) among the groups in the same period of analysis. The line represents no significant differences (*p* > 0.05) among the groups in same period of analysis. The different capital letters (Control: A and B; Exposed: C and D; Injury: E, F, and G; LLLT: H and I; F1: J, K, and L; LLLT + F1: M, N, and O) represent significant differences (*p* < 0.05) among the same group in different periods of analysis.

**Figure 3 ijms-24-14031-f003:**
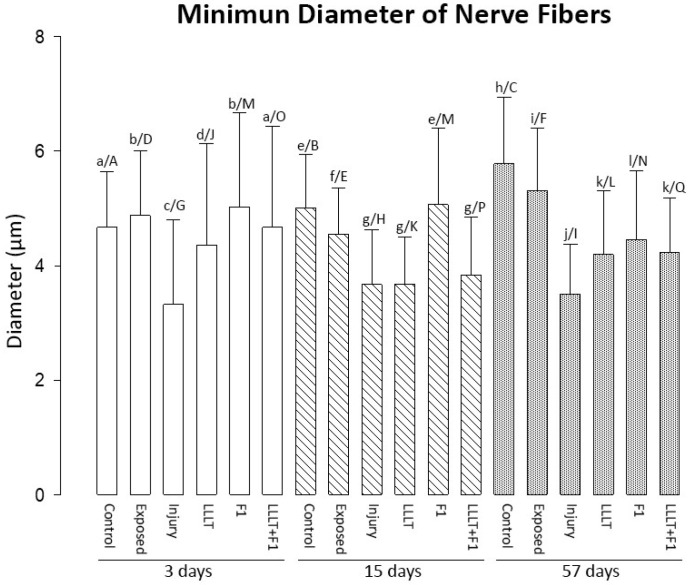
Minimum nerve fiber diameter after 3, 15, and 57 days of nerve injury. The different lower-case letters (3 days: a, b, c, and d; 15 days: e, f, and g; 57 days: h, i, j, k, and l) represent significant differences (*p* < 0.05) among the groups in the same period of analysis. The different capital letters (Control: A, B, and C; Exposed: D, E, and F; Injury: G, H, and I; LLLT: J, K, and L; F1: M and N; LLLT + F1: O, P, and Q) represent significant differences (*p* < 0.05) among the same group in different periods of analysis.

**Figure 4 ijms-24-14031-f004:**
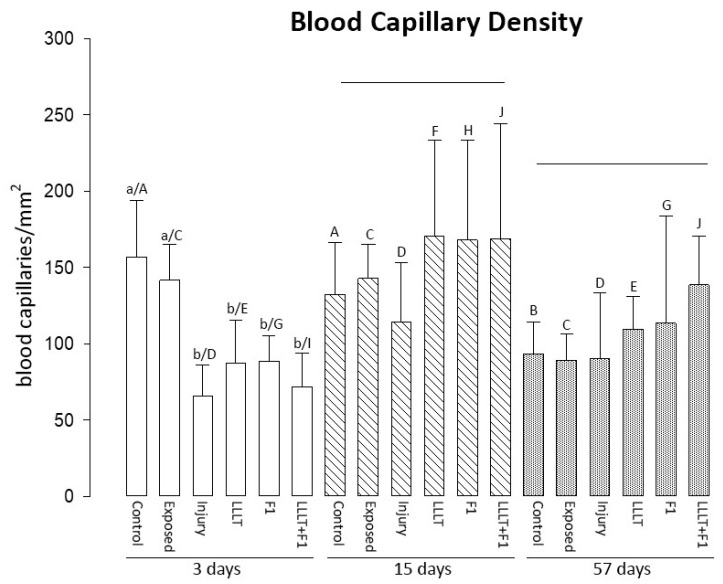
Capillary Density after 3, 15, and 57 days of nerve injury. The different lower-case letters (3 days: a and b) represent significant differences (*p* < 0.05) among the groups in the same period of analysis. The lines represent no significant differences (*p* > 0.05) among the groups in the same period of analysis. The different capital letters (Control: A and B; Exposed: C; Injury: D; LLLT: E and F; F1: G and H; LLLT + F1: I and J) represent significant differences (*p* < 0.05) among the same group in different periods of analysis.

**Figure 5 ijms-24-14031-f005:**
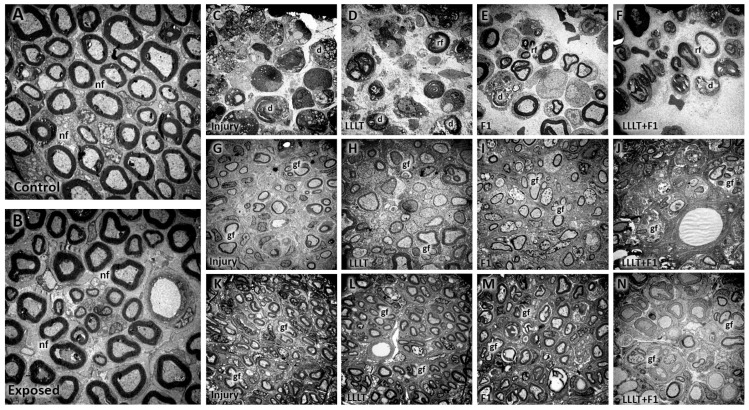
Ultrastructural analysis using TEM. (**A**) Control group and (**B**) Exposed group, revealing normal nerve fibers (nf) 3 days after nerve injury. (**C**) Injury group. (**D**) LLLT group. (**E**) F1 group and (**F**) LLLT + F1 group, showing degeneration regions (d) and nerve fibers reduced (rf) in size and quantity 15 days after nerve injury. (**G**) Injury group. (**H**) LLLT group. (**I**) F1 group and (**J**) LLLT + F1 group, revealing some grouping of nerve fibers (gf) reduced in size 57 days after nerve injury. (**K**) Injury group. (**L**) LLLT group. (**M**) F1 group and (**N**) LLLT + F1 group, showing grouping of higher number of nerve fibers (gf) reduced in size. Magnification: A–N: ×2000.

**Figure 6 ijms-24-14031-f006:**
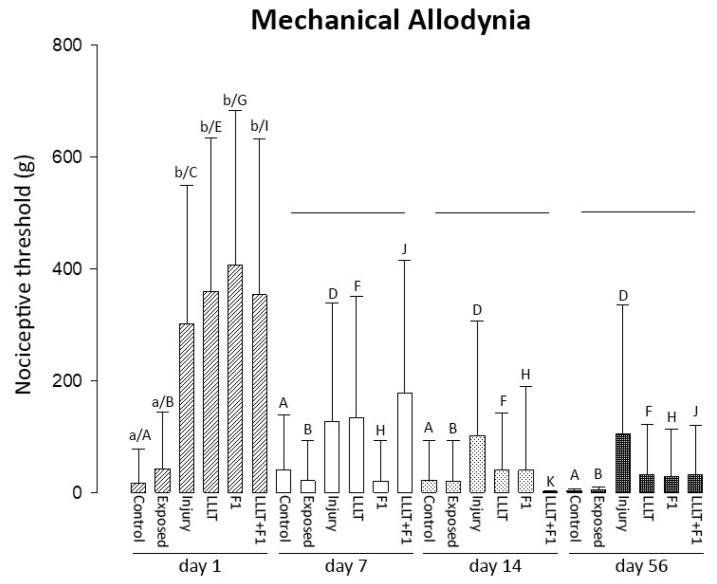
Mechanical allodynia after 1, 7, 14, and 56 day(s) of nerve injury. The different lower-case letters (day 1: a and b) represent significant differences (*p* < 0.05) among the groups in the same period of analysis. The lines represent no significant differences (*p* > 0.05) among the groups in the same period of analysis. The different capital letters (Control: A; Exposed: B; Injury: C and D; LLLT: E and F; F1: G and H; LLLT + F1: I and J) represent significant differences (*p* < 0.05) among the same group in different periods of analysis.

**Figure 7 ijms-24-14031-f007:**
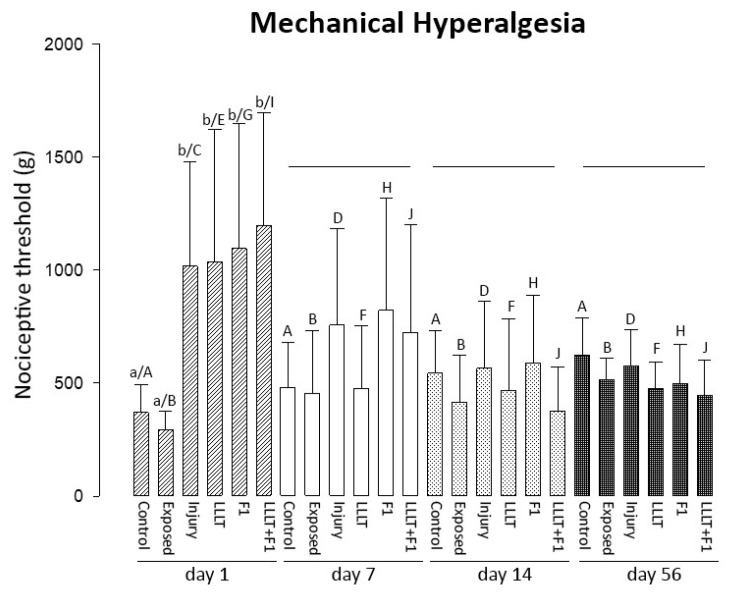
Mechanical hyperalgesia after 1, 7, 14, and 56 day(s) of nerve injury. The different lower-case letters (day 1: a and b) represent significant differences (*p* < 0.05) among the groups in the same period of analysis. The lines represent no significant differences (*p* > 0.05) among the groups in the same period of analysis. The different capital letters (Control: A; Exposed: B; Injury: C and D; LLLT: E and F; F1: G and H; LLLT + F1: I and J) represent significant differences (*p* < 0.05) among the same group in different periods of analysis.

**Figure 8 ijms-24-14031-f008:**
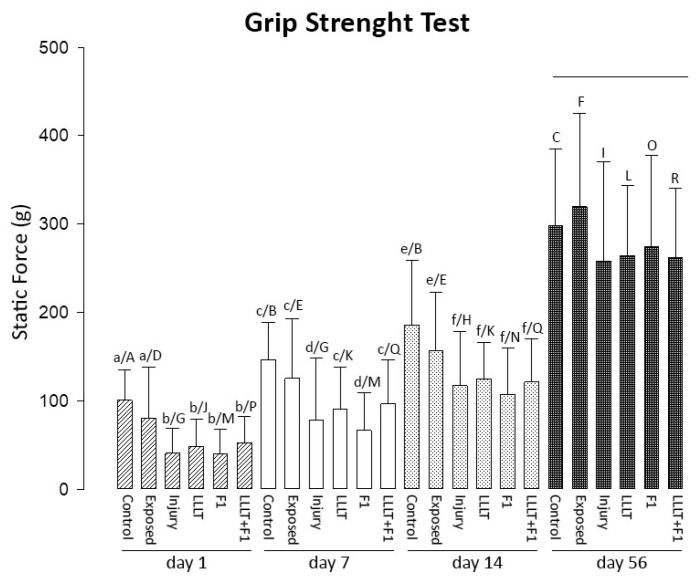
Grip Strength test after 1, 7, 14, and 56 day(s) of nerve injury. The different lower-case letters (day 1: a and b, day 7: c and d; day 14: e and f) represent significant differences (*p* < 0.05) among the groups in the same period of analysis. The lines represent no significant differences (*p* > 0.05) among the groups in the same period of analysis. The different capital letters (Control: A, B, and C; Exposed: D, E, and F; Injury: G, H, and I; LLLT: J, K, and L; F1: M, N, and O; LLLT + F1: P, Q, and R) represent significant differences (*p* < 0.05) among the same group in different periods of analysis.

**Figure 9 ijms-24-14031-f009:**
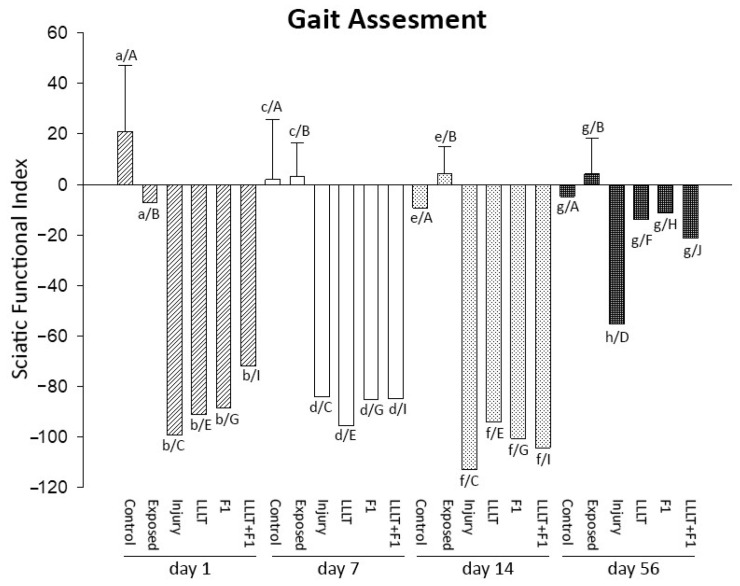
Gait analysis after 1, 7, 14, and 56 day(s) of nerve injury. The different lower-case letters (day 1: a and b, day 7: c and d; day 14: e and f; day 56: g and h) represent significant differences (*p* < 0.05) among the groups in the same period of analysis. The different capital letters (Control: A; Exposed: B; Injury: C and D; LLLT: E and F; F1: G and H; LLLT + F1: I and J) represent significant differences (*p* < 0.05) among the same group in different periods of analysis.

**Figure 10 ijms-24-14031-f010:**
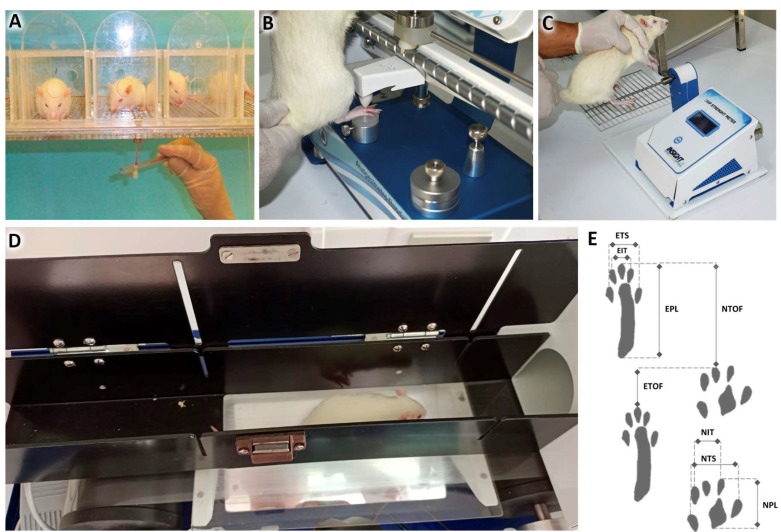
Functional analysis. (**A**) Mechanical allodynia test using von Frey hair. (**B**) Hyperalgesia test using Randall–Selitto apparatus. (**C**) Grip strength test using grip strength meter apparatus. (**D**) Gait test using gait analyzer apparatus. (**E**) Sciatic functional index measurements, where ETS stands for: experimental side total extension, EIT: Experimental side intermediate fingers, EPL: Experimental side print leght. NTS: normal side total extension, NIT: Normal side intermediate fingers and NPL: normal side print leght [62].

**Table 1 ijms-24-14031-t001:** Low-level laser parameters used.

Parameter	Value
Output power	30 mW
Power density	0.75 W/cm^2^
Energy density	15 J/cm^2^
Wavelength	780 nm
Application time (per point)	20 s
Number of application points	3
Wave type	Continuous wave
Beam direction	Perpendicular to tissue
Dose per treatment spot (per point)	0.6 J
Spot area of application	0.04 cm^2^
Number of irradiation sessions	6 (alternate days)

**Table 2 ijms-24-14031-t002:** Experimental protocols applied per group of animals.

Group	Experimental Protocol Applied
Control	Anesthetized animals were subjected to trichotomy in the left hind paw and kept in lateral decubitus for 10 min (*n* = 18).
Exposed	The uninjured exposed sciatic nerve was placed on the lesion support for 10 min (*n* = 18).
Injury	The sciatic nerve was injured via crushing (150 N/15 kgf, 10 min) and repositioned without treatment (*n* = 18).
LLLT	The injured sciatic nerve was repositioned and, after two days, irradiated with LLLT (15 J/cm^2^, 780 nm) (*n* = 18).
F1	The injured sciatic nerve was repositioned, and we subsequently applied hyaluronic acid with purified natural latex protein (0.1% concentration) around the nerve at the injury site (*n* = 18).
LLLT + F1	The injured sciatic nerve was repositioned, and we subsequently applied F1 (0.1%). After 2 days, the animals were irradiated with LLLT (15 J/cm^2^, 780 nm) (*n* = 18).

## Data Availability

The data that support the findings of this report are available from the corresponding author, [F.J.D.], upon reasonable request.

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
