# Peer review of "Effects of Low-Level Laser Therapy and Purified Natural Latex (Hevea brasiliensis) Protein on Injured Sciatic Nerve in Rodents: Morpho-Functional Analysis"

_ijms, 2023, doi:10.3390/ijms241814031_

Round 1
Reviewer 1 Report
I read the manuscript with interest. The study concerned the intriguing effects of Low-level laser therapy (LLLT) and the purified natural latex protein (Hevea brasiliensis, F1 protein) on the morpho-function of sciatic nerve crush injuries in rats. Overall, I find the paper to be of great quality, and I appreciate the effort put into this research.
I suggest the following title: Effects of Low-Level Laser Therapy and Purified Natural Latex (Hevea Brasiliensis) Protein on Injured Sciatic Nerve in Rodents. I leave this to the authors and editor to decide.
Lastly, I would like to highlight a minor structural consideration. It is evident that the results section precedes the method section in the current arrangement. Placing the method section before the results would provide a smoother flow for readers to better understand the experimental context for the outcomes.
In terms of content, I am impressed by the comprehensive approach and thorough analysis presented in the study. The utilization of various assessments at different time-points adds depth to the research, and the discussion of sensory and motor analyses offers valuable insights into the observed effects. I commend the authors for addressing the morphometric and ultrastructural parameters, contributing to a well-rounded investigation.
In conclusion, I think the paper holds substantial promise and contributes valuable insights to the field.
Here are some additional comments.
The readability of Table 1 is poor due to the interquartile range being presented with a Q twice after every point estimate. At the top of the table, consider writing IQR for the interquartile range just once. Remember to explain the abbreviations.
The figures on the right side of page 8 are difficult to read due to their small size.
In Table 3, "Power" should be "Output power" and "Intensity" should be "Power density". "Deposited Energy (per point)" should be renamed into, e.g. "Dose per treatment spot". These are the common terms.
Author Response
Comments and Suggestions for Authors
I read the manuscript with interest. The study concerned the intriguing effects of Low-level laser therapy (LLLT) and the purified natural latex protein (Hevea brasiliensis, F1 protein) on the morpho-function of sciatic nerve crush injuries in rats. Overall, I find the paper to be of great quality, and I appreciate the effort put into this research.
Reply: The authors would like to thank you for your comments on the manuscript. All your observations were accepted and included in the revised manuscript.
I suggest the following title: Effects of Low-Level Laser Therapy and Purified Natural Latex (Hevea Brasiliensis) Protein on Injured Sciatic Nerve in Rodents. I leave this to the authors and editor to decide.
Reply: Thanks for commenting on the title of the article. The authors decided to modify it to “Low-Level Laser and Natural Latex (Hevea Brasiliensis) Purified Protein on the Injured Sciatic Nerve in Rodents: Morpho-Functional Analysis” keeping the term “morpho-functional analysis” to help differentiate it from other studies of our research group.
Lastly, I would like to highlight a minor structural consideration. It is evident that the results section precedes the method section in the current arrangement. Placing the method section before the results would provide a smoother flow for readers to better understand the experimental context for the outcomes.
Reply: We appreciate your comment on the structure of the article, and we share the same opinion. However, we must follow the journal rules that request that the results and discussion sections be presented before the materials and methods.
In terms of content, I am impressed by the comprehensive approach and thorough analysis presented in the study. The utilization of various assessments at different time-points adds depth to the research, and the discussion of sensory and motor analyses offers valuable insights into the observed effects. I commend the authors for addressing the morphometric and ultrastructural parameters, contributing to a well-rounded investigation.
In conclusion, I think the paper holds substantial promise and contributes valuable insights to the field.
Reply: Reply: Thank you for your observation and good evaluation of our manuscript.
Here are some additional comments.
The readability of Table 1 is poor due to the interquartile range being presented with a Q twice after every point estimate. At the top of the table, consider writing IQR for the interquartile range just once. Remember to explain the abbreviations.
Reply: Thanks for your observation, based on the comment of another reviewer, the authors decided to leave the presentation of the results in graph format only. Because, according to the comment, presenting graphs and tables of the same data could confuse readers.
The figures on the right side of page 8 are difficult to read due to their small size.
Reply: We appreciate the observation. The figure on page 8 was modified based on the request of another reviewer. Thus, the graphs are being presented in a larger size to improve the reading of the study.
In Table 3, "Power" should be "Output power" and "Intensity" should be "Power density". "Deposited Energy (per point)" should be renamed into, e.g. "Dose per treatment spot". These are the common terms.
Reply: We appreciate your suggestions. Table 1 (formerly Table 3) has been modified as indicated.

Reviewer 2 Report
The study under review is indeed interesting and possesses merit. However, several key issues need to be addressed to enhance the rigor and readability of the manuscript. Here are my specific comments and recommendations:
-
Placebo Control for F1: The absence of a placebo control for F1 when comparing LLLT + F1 with LLLT alone raises concerns about the reliability of the findings. The injection solution itself could introduce an unknown variable, thereby affecting the overall results. I recommend introducing a placebo control for a more robust comparison.
-
Presentation of Results: While the results are systematically presented, the section is lengthy and requires multiple readings to fully grasp the implications. I suggest condensing the results section for easier comprehension. Graphical representation could be a helpful addition; for instance, plotting time on the x-axis and variables like fiber density on the y-axis for all groups could make the differences more immediately clear.
-
Statistical Analysis: Employing one-way ANOVA to compare the same variables at different time points is not appropriate. A two-way ANOVA for repeated measurements, or an equivalent non-parametric test, would be more suitable for this type of data.
-
Duplication of Results: The results are redundantly presented in both figures and tables. This is unnecessary and could lead to reader confusion. Please choose one format for presenting each result to enhance clarity.
-
Figures and Images: The manuscript contains numerous images, many of which are well-executed. However, to streamline the content, consider relocating some of these images to the supplementary materials.
-
Discussion's Opening Paragraph: The first paragraph of the discussion section should succinctly summarize the principal findings of the study to orient the reader.
-
Discussion Language and Context: The current discussion is technically dense and largely descriptive. I recommend rewriting this section to improve fluency and to better situate your findings within the context of existing research and potential future studies.
- Inflammatory markers: Did you check for possible inflammation and inflammatory cells in the nerve sections?
I believe that addressing these issues will significantly improve the manuscript and make it more suitable for publication.
The language is cumbersome and very technical with grammatical mistakes.
Author Response
The study under review is indeed interesting and possesses merit. However, several key issues need to be addressed to enhance the rigor and readability of the manuscript.
Reply: The authors would like to thank you for your review of the manuscript. All suggestions were accepted, and the corresponding changes were included in the revised version of the manuscript.
Here are my specific comments and recommendations:
- Placebo Control for F1: The absence of a placebo control for F1 when comparing LLLT + F1 with LLLT alone raises concerns about the reliability of the findings. The injection solution itself could introduce an unknown variable, thereby affecting the overall results. I recommend introducing a placebo control for a more robust comparison.
Reply: We appreciate your comment regarding the use of a placebo to analyze the effects of the F1 protein. The authors decided not to include this group in this study because they had already analyzed the F1 protein vehicle, hyaluronic acid, in a previous study (DOI: 10.3109/08977194.2014.952727) which revealed that the vehicle was a factor that worked better when associated with protein F1. Furthermore, this would increase the number of study groups and animals used in research where the focus was on analyzing LLLT and the F1 protein and their association. Anyway, the idea of analyzing only the effect of the vehicle associated with the protein and the LLLT could be carried out in future studies.
- Presentation of Results: While the results are systematically presented, the section is lengthy and requires multiple readings to fully grasp the implications. I suggest condensing the results section for easier comprehension. Graphical representation could be a helpful addition; for instance, plotting time on the x-axis and variables like fiber density on the y-axis for all groups could make the differences more immediately clear.
Reply: Thank you for your observation. In fact, the presentation of the amount of data from this research was an initial “problem”. Which we recognize was not resolved in the best way in the initial version of the manuscript. The entire section of results was revised and the presentation of data was modified to favor understanding of the study. Presenting a smaller amount of images in a more organized way, which favors comparisons of the study groups and analyzed times. In addition to organizing the results in a smaller number of graphs, which also favors, in our opinion, the understanding of the study.
- Statistical Analysis: Employing one-way ANOVA to compare the same variables at different time points is not appropriate. A two-way ANOVA for repeated measurements, or an equivalent non-parametric test, would be more suitable for this type of data.
Reply: Thank you for your observation. In fact, the use of two-way ANOVA was more suitable for the study. Thus, all statistical analysis and graphical presentations were modified based on your suggestion. We recognize that the graphical presentation of the quantitative data analyzes is now much more understandable.
- Duplication of Results: The results are redundantly presented in both figures and tables. This is unnecessary and could lead to reader confusion. Please choose one format for presenting each result to enhance clarity.
Reply: Thank you for your observation and we fully agree. Thus, only the presentation in the form of graphics was presented in the revised version of the manuscript.
- Figures and Images: The manuscript contains numerous images, many of which are well-executed. However, to streamline the content, consider relocating some of these images to the supplementary materials.
Reply: Thank you for your observation. The images were reorganized, eliminating “repeated” images from the non-injured groups. The way they are arranged in the revised version will allow a better comparison of protocols and analysis times.
- Discussion's Opening Paragraph: The first paragraph of the discussion section should succinctly summarize the principal findings of the study to orient the reader.
Reply: Thank you for your observation. First of all, I would like to apologize for the carelessness of having left an initial paragraph of the discussion with the journal's guidelines. This paragraph was deleted and a new opening paragraph with the main findings of the study was included at the beginning of the discussion.
- Discussion Language and Context: The current discussion is technically dense and largely descriptive. I recommend rewriting this section to improve fluency and to better situate your findings within the context of existing research and potential future studies.
Reply: Thank you for your observation. The entire discussion has been revised and modified for readability.
- Inflammatory markers: Did you check for possible inflammation and inflammatory cells in the nerve sections?
Reply: Thank you for your observation, in the present study we did not focus on the analysis of cells and inflammatory markers. The future intention of our research group is to also focus on analyzes of inflammatory cells and inflammatory markers such as pro-inflammatory interleukins. In a previous study, the presence of mast cells was noted under the transmission electron microscope (DOI: 10.1371/journal.pone.0210211) and it is known that the analysis of these and markers is of fundamental importance in nerve regeneration. Thus, this information was included in the limitations of the present study.

Round 2
Reviewer 2 Report
The authors satisfactorily addressed my comment.s